# Ostracod Fauna: Eyewitness to Fifty Years of Anthropic Impact in the Gulf of Trieste. A Potential Key to the Future Evolution of Urban Ecosystems

**Gianguido Salvi** [1,*] , **Alessandro Acquavita** [2] , **Massimo Celio** [2] , **Saul Ciriaco** [3] , **Stefano Cirilli** [1] , **Michele Fernetti** [1] and **Nevio Pugliese** [1]

[1] Department of Mathematics and Geosciences, University of Trieste, Via E.Weiss 2, 34127 Trieste, Italy; cirilli@units.it (S.C.); fernetti@units.it (M.F.); npugliese712@gmail.com (N.P.)

[2] ARPA (Regional Environmental Protection Agency, Department of Trieste), Via La Marmora 13, 34139 Trieste, Italy; alessandro.acquavita@arpa.fvg.it (A.A.); massimo.celio@arpa.fvg.it (M.C.)

[3] WWF Miramare MPA, Viale Miramare 349, 34151 Trieste, Italy; saul@riservamarinamiramare.it

\* Correspondence: gsalvi@units.it

**Abstract:** For the first time, the distribution and modifications of living ostracod associations present in the Gulf of Trieste (GoT) in relation to alterations caused by human activity in the last 20 years were investigated. The results were compared with the main physicochemical parameters (especially nitrogen and phosphorus) measured over the same period, which can lead to a general decrease in environmental quality. For a more in-depth analysis of the changes recorded by ostracods in the last 50 years, a period in which eutrophication and anoxia increased, we revisited the study carried out by Masoli in the GoT in 1967. The results obtained made it possible to verify how, over the last 20 years, ostracod assemblages have suffered a decrease both qualitatively and quantitatively. Most of the species recovered show characteristics of opportunism and tolerance to environmentally stressful conditions, high organic matter concentrations, and oxygen deficiency. The ostracods analyzed in 1967 showed similar results with few dominant opportunistic species. We verified how ostracods recorded in GoT, similar to Mollusks and Foraminifera, have been impaired by the possible environmental crisis linked to the recurrence of mucilage and hypoxic events documented for the GoT in the last 50 years. Finally, a comparison with the best environmental conditions found in the Marine Nature Reserve of Miramare (MPA) allowed us to emphasize the important role of protected areas to avoid loss of biodiversity due to urbanization.

**Keywords:** urbanization; Gulf of Trieste; ostracods; nutrients; environmental stress; marine nature reserve

## 1. Introduction

Urbanization is one of the main causes of species extinction [1]. The expansion and growth of anthropic activities decrease biological diversity because the same "urban-adaptable" species become widespread and locally abundant across several ecosystems [2,3]. Thus, a great deal of research has been focused on human influence generating ecological degradation from coastal urbanization to pollution and eutrophication [4,5]; these fundamental changes have taken place with the evolution of freshwater systems and fluxes during the last century, with particular impact seen over the last 50 years [5,6]. More specifically, drastically increased urbanization is currently one of the main sources of anthropogenic impact [7,8], often exerting significant pressure on coastal ecological systems [9].

The need to start efficient mitigation activities to assess and manage the negative impacts of urbanization on natural habitats (i.e., to reduce urban footprints and to preserve habitats in urbanized

areas) has highlighted the importance of investigating recent anthropic impact and the response of contiguous marine ecosystems. In this context, the Gulf of Trieste (GoT) represents a suitable area to investigate recent human impact on a coastal marine environment, as this area has been densely urbanized over a long period and subject to agricultural activities in the inland Friulian Plain. The presence of wastewater discharge and riverine inputs have caused episodes of anoxia [10] similar to those observed in other coastal areas [5,11–13].

The GoT is characterized by several sources of pressure such as industrial sites, two busy harbors (Trieste and Monfalcone), intense ship traffic, aquaculture, mussel farming, and tourism facilities. The effects of anthropogenic disturbances in the GoT were investigated taking into account the presence and distribution of geochemical pollutants and biological markers and by using an integrated approach [14–16]. Adami et al. [17,18] studied the close relationship between biometric parameters for sediment dwelling organisms and the heavy metal content and assessed the relevance of anthropogenic Cu, Pb, Zn, and Cd as a factor of pollution, conditioning benthic life in the harbor of Trieste.

Barbieri et al. [19], investigating surface sediments near urban and industrial sewage discharges, pointed out the noxious, recent anthropogenic pressure on the benthic environment related to the presence of anomalously high concentrations of copper and zinc. The experimental evidence suggested the need for stricter control of heavy metal contents in sewage before diffusion to the open sea in order to avoid further compromise of marine life at the seawater/sediment interface and, indirectly, human health in coastal communities.

Moreover, the GoT is part of one of the most polluted areas in the Mediterranean. Apart from issues with mercury (Hg), the GoT is also subject to industrial and sewage pollution [20]. Due to deteriorating water quality in the Gulf, there has been great concern that Hg could be remobilized from sediment to the water column as well as could enhance methylation rates which may consequently increase already elevated Hg levels in aquatic organisms. Hg exhibited higher concentrations in the surface layer in the area in front of the river plumes. Higher bottom concentrations of dissolved Hg observed at some stations were likely due to remobilization from sediments, including resuspension and benthic recycling [21,22].

Acquavita et al. and Petranich et al. [23,24], analyzing several coastal sites and harbor areas in the GoT, demonstrated how port areas or marinas, in particular, were highly impacted by various contaminants (trace metals, polycyclic aromatic hydrocarbons, polychlorobiphenyls, organotin compounds, etc.).

On this basis, a restricted area of the GoT was identified as heavily polluted and defined as a Site of National Interest (SIN) (d.lgs. 22/97; d.lgs. 152/2006) which required cleanup [25].

Recent analysis investigated the relationship between benthic communities and anthropization in areas of growing urbanization [4,26]. The analysis of microbenthic communities found where wastewater enters the GoT indicates a degree of environmental stress due to imbalance, showing how waste treatment has been effective at controlling the adverse effects of urban discharges [27]. Tomašových et al. [28] found that production of the opportunistic bivalve *Corbula gibba* strongly fluctuated over the past few centuries and suggested that intervals with higher frequencies of hypoxia were not exclusively driven by human-induced enrichment in the 20th century. The long-term response of whole macrobenthic communities to natural or anthropogenic impacts in the GoT remains largely unknown.

Gallmetzer et al. [4] recorded a decline in some molluscan species in the second half of the twentieth century as well as an associated increase in some opportunistic species as a result of an increase in fishing and hypoxia phenomena.

Recent studies on foraminifera communities [29,30] highlighted the response of this taxon to the various stimuli linked to anthropogenic factors and, consequently, to the degree of environmental quality, directly related to levels of pollutants and/or a trophic state (i.e., anoxia/hypoxia phenomena).

A sensitive species, ostracods are small crustaceans (from 500 to 1500 μm in length) which occupy all aquatic environments (from the deep sea to inland freshwater ponds and wetland to terrestrial

environments) and are capable of secreting calcareous carapaces. Their distribution is controlled by hydrological, biological, and sedimentological features. These organisms are useful for environmental characterization on a local/regional scale and can indicate water depth, salinity, temperature, and other ecological factors. Numerous authors have reported the use of ostracods for environmental and paleoenvironmental studies as sentinels of anthropic impact and associated pollution and hypoxia phenomena [31,32].

The effects of eutrophication on ostracod associations due to anthropic activities was studied by [33,34]. Cronin and Vann [33], through the analysis of sedimentary records collected in the Patuxent Estuary and Chesapeake Bay ecosystems, verified the migration of *Loxoconcha* spp. from the deep channel into shallower water along the flanks of the bay due to a combination of increased nitrate loading and fertilizer use. It also seems probable that the four- to five-fold increase in sedimentation due to agricultural and timber activity may have contributed to an increased natural nutrient load, likely fueling the early periods (1700–1900) of hypoxia prior to widespread fertilizer use. Twentieth-century anoxia worsened in the late 1930s–1940s and again around 1970, reaching unprecedented levels in the past few decades. Similar effects were found in the Gernika estuary (southern Bay of Biscay) during summer periods waters [34].

The distribution of ostracods was investigated at sites with different pollution levels on the eastern coast of Amurskii Bay within the limits of Vladivostok City and in the Gulf of Izmir. It was found that pollution reduces species diversity and causes changes in community structure and, eventually, the total extinction of ostracods. The study showed that *Xestoleberis* spp. is the most resistant to anthropogenic pressure and could survive in areas where all other ostracod species have already become extinct [35,36].

Analysis of cores collected in the Odiel Estuary (SW Spain) allows one to delimit the recent evolution of this zone during the past decades and the influence of natural and anthropogenic factors on the distribution of ostracods. In the upper estuary, coinciding factors such as acid waters, prolonged subaerial exposure, and coarse sediments may explain the absence or disappearance of ostracod assemblages during the industrial period (1966–1985) in the major part of this area. In the lower estuary, sedimentary evolution and industrial wastes are the main factors influencing both distribution and trends of populations. Finally, the main changes observed in the marine estuary are due to sedimentary effects of the construction of two banks and dredging of the main estuarine channel [37–39].

Temporal changes in ostracods observed in sediment cores from Hiroshima and Osaka Bay (Japan) provide valuable information regarding the influence of anthropogenic pollution. Results suggest that industrialization and anthropogenic pollution caused a decrease in ostracod density and the homogenization of ostracod assemblages in Hiroshima Bay [40]. Urbanization-induced eutrophication recorded on a metazoan benthic community in Osaka Bay suggests that the total abundance of ostracod decreased in the inner bay, likely due to bottom-water hypoxia by eutrophication. The variation in species composition within the two bays may have decreased because of the effects of eutrophication, i.e., the dominance of species that prefer food-rich environments [41,42].

Finally, some studies have focused on the response of these organisms to coastal human impact in terms of water pollution by industrial, agricultural, and military processes, sewage and the resulting eutrophication that may lead to hypoxia and, in extreme cases, may induce anoxia [43–45]. In this context, ostracods are usually intolerant of hypoxia and respond by a reduction in diversity and richness, and in some cases, populations become monospecific [5,46,47].

In this work, we examined the ostracod population in the GoT over the last 20 years by applying an integrated, multidisciplinary approach to reconstruct and evaluate the recent history of the impact of urbanization on communities of these small crustaceans. The main physicochemical parameters commonly employed to define the trophic state (i.e., nutrients, chlorophyll *a*, temperature, and salinity) were also taken into consideration to check the relationships with changes in population. In addition, we compared the more recent data with the results published by Masoli [48] to verify changes in association (appearance and disappearance) recorded over a time span of 50 years.

The research also represents the first attempt at a large time analysis of the possible relationship between ostracod assemblages inside a selected area of the GoT and modification in a close urban context.

The evaluation of their modifications and adaptations over time could represent a crucial factor in setting out future recovery actions and could aid in implementing a sustainable urban plan, often invoked as a "win–win–win" scenario to optimize economic, environmental, and social goals [49].

## 2. Materials and Methods

### 2.1. Study Area

The GoT is located in the northernmost part of the Adriatic Sea (Italy). It is an epicontinental semi-enclosed shelf basin covering an area of approximately 500 km$^2$ with a maximum water depth of 25 m (average depth 17 m) and characterized by a very low bathymetric gradient. It is affected by the significant contribution of continental waters coming from the Italian and Istrian regions [50]. The primary freshwater input is represented by the Isonzo/Soča River (annual flow rate of 82 m$^3$ s$^{-1}$), and the contribution of several minor rivers (Timavo/Reka, Rosandra/Glinščica, Ospo/Osp, Rižana, Badaševica, Drnica, and Dragonja) can be considered negligible or having only local effects. Water circulation is driven by the interplay of various forcing factors: the general circulation of the Adriatic Sea, winds (particularly the dominant Bora, N-NE direction), and buoyancy fluxes together with tides. The GoT represents a site of shelf dense water formation that contributes to the North Adriatic Deep Water.

The sediment texture varies from medium to fine sands along the coastline and the delta of the Isonzo and Tagliamento Rivers to muds in the mid-Gulf and sandy sediments in the western open part of the GoT. Carbonate sediments dominate the sediments near the river's mouth [51,52].

The GoT is a suitable site to study anthropic impact since, in spite of its relatively small extension; it hosts two of the largest cargo shipping ports in the Adriatic Sea (Trieste and Koper). This coastal area is affected by many potential sources of organic and inorganic pollutants, discharged not only by rivers but also by sewers, industrial developments, and harbor-related activities including an oil-pipeline terminal [53,54]. Moreover, the site has been recognized as an area where particular conditions related to inputs of fluvial sediments or to meteo-marine conditions led to significant algal productions and blooms, resulting in eutrophication and the subsequent hypoxic/anoxic conditions at the bottom, at least until the mid-1980s [55].

### 2.2. Experimental Site

The experimental site is located in the GoT (13°37′ E to 13°44′ E and 45°41′ N to 45°44′ N). Within this area, 44 samples were collected during two different summer cruises conducted in 2004–2005 and 2013–2017, hereafter referred to as GTCrB and GTCrC, respectively (Figure 1). Sampling was performed using a five-liter Van Veen grab and a KC Haps bottom corer characterized by a sample area of 0.013 m$^2$ with an effective depth penetration of 10 cm.

In order to assess the changes which have occurred in the GoT since 1967, we applied Geographic Information System (GIS) to reconstruct the exact georeferenced location of the samples collected by the "Istituto di Geologia e Paleontologia" (University of Trieste) during two summer cruises conducted in 1965 and 1966, respectively (site GTCrA), and compared the qualitative analyses of ostracods reported by Masoli [48] with those of ostracods recovered in 2004 and 2017 (Figure 1).

A schematic presentation of the methodology used in this study is shown in Figure 2.

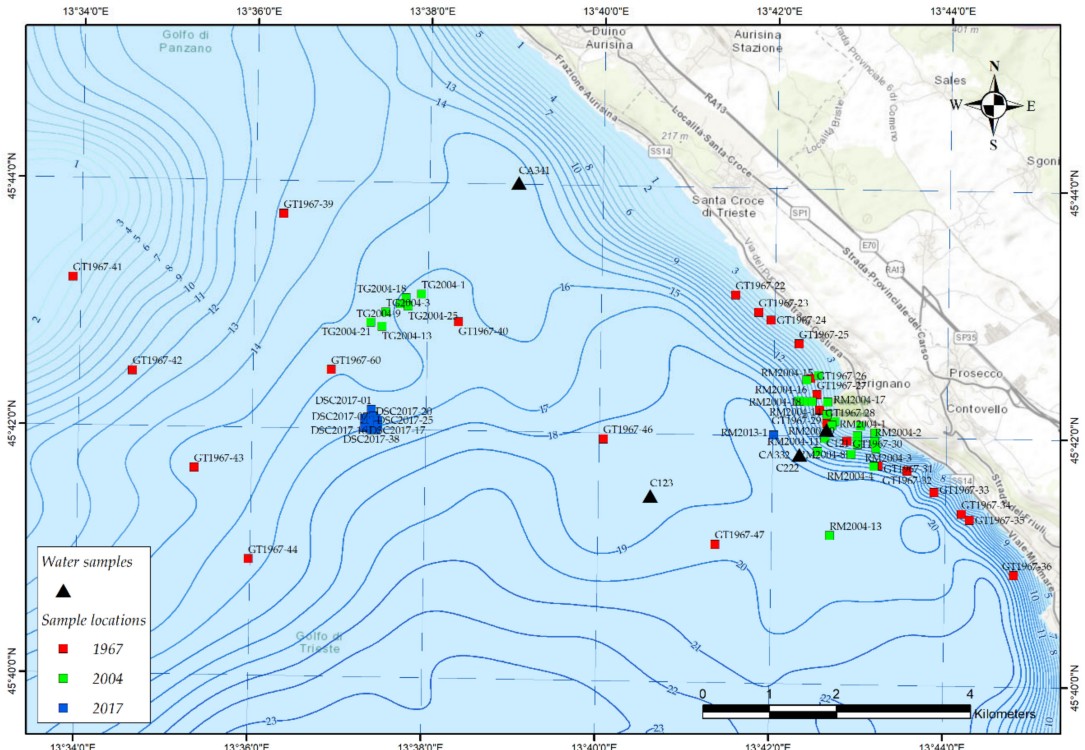

**Figure 1.** Location of the study area and sampling stations: the authors' elaboration from Esri, HERE, Garmin, Intermap, increment P Corp., GEBCO, USGS, FAO, NPS, NRCAN, GeoBase, IGN, Kadaster NL, Ordinance Survey, Esri Japan, METI, Esri China (Hong Kong), OpenStreetMap contributors, and the GIS User Community. Bathymetric datasets come from the OGS SNAP data repository (source: http://doi.org/cpz2).

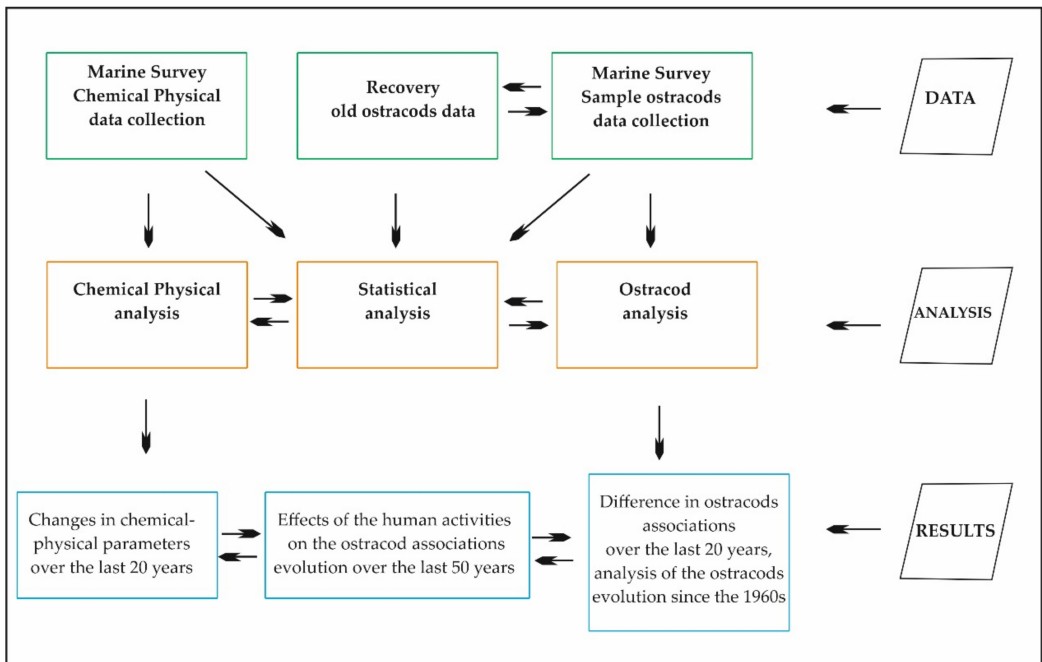

**Figure 2.** Schematic presentation of the methodology used in this study: the authors' elaboration.

The collected sediments were wet-sieved using a 63-μm mesh, dried, and weighed to determine the content of the sandy fraction. The total sandy sediments were used to collect all live benthic specimens, representative of the environmental conditions, and to avoid problems related to the

potential presence of the reworked fauna originating from deposits created in connection with the Early Holocene marine transgressions and the Holocene climatic optimum as indicated by Uffenorde [56] in the eastern part of the northern Adriatic Sea. Ostracod analysis took into account monographs and papers from the Mediterranean literature [57–59]. Particular attention was paid to the papers concerning the northernmost sector of the Adriatic Sea corresponding to the GoT [48,60,61].

A similarity percentage (SIMPER) analysis defined the main taxa responsible for the differences between groups. These analyses were based on the Bray–Curtis dissimilarity, a measure that does not take into account the absence of species but focuses on the composition of assemblage and the relative abundances of taxa [62].

To characterize the biodiversity of assemblages, two faunal parameters were calculated: (1) species diversity (S), the number of species in each sample, and (2) the Shannon–Weaver index (H), a measure of entropy that takes into considerations the distribution of taxa among the total individuals [63] (Table 1).

**Table 1.** Species Richness (S), Total Density (i.e., the Number of Individuals in Each Sample), and the Shannon-Weaver Diversity Index Calculated for Benthic Living Ostracods in Each Sample.

| Samples | Taxa_S | Individuals | Shannon_H | Samples | Taxa_S | Individuals | Shannon_H |
|---------|--------|-------------|-----------|---------|--------|-------------|-----------|
| TG2004-1 | 13 | 41 | 2.4 | RM2004-16 | 11 | 77 | 2.0 |
| TG2004-3 | 8 | 16 | 2.0 | RM2004-17 | 5 | 21 | 1.4 |
| TG2004-13 | 19 | 53 | 2.7 | RM2004-19 | 9 | 68 | 1.3 |
| TG2004-18 | 14 | 57 | 2.2 | RM2004-20 | 10 | 93 | 1.8 |
| TG2004-21 | 14 | 110 | 1.9 | DSC2017-01 | 6 | 9 | 1.7 |
| TG2004-25 | 14 | 56 | 2.2 | DSC2017-07 | 5 | 41 | 1.0 |
| RM2004-1 | 8 | 16 | 1.6 | DSC2017-10 | 7 | 8 | 1.9 |
| RM2004-2 | 9 | 45 | 1.5 | DSC2017-12 | 7 | 61 | 1.2 |
| RM2004-3 | 11 | 76 | 2.1 | DSC2017-13 | 8 | 31 | 1.3 |
| RM2004-4 | 11 | 84 | 2.0 | DSC2017-16 | 7 | 26 | 1.3 |
| RM2004-5 | 5 | 6 | 1.7 | DSC2017-17 | 5 | 13 | 1.4 |
| RM2004-6 | 14 | 69 | 1.6 | DSC2017-18 | 3 | 9 | 0.7 |
| RM2004-7 | 6 | 44 | 1.2 | DSC2017-19 | 4 | 15 | 0.7 |
| RM2004-8 | 6 | 38 | 1.5 | DSC2017-20 | 4 | 7 | 1.3 |
| RM2004-9 | 14 | 64 | 1.9 | DSC2017-21 | 3 | 4 | 1.0 |
| RM2004-10 | 13 | 96 | 1.8 | DSC2017-25 | 8 | 24 | 1.7 |
| RM2004-11 | 11 | 99 | 1.9 | DSC2017-32 | 4 | 4 | 1.4 |
| RM2004-12 | 10 | 108 | 1.9 | DSC2017-34 | 6 | 18 | 1.4 |
| RM2004-13 | 8 | 26 | 1.8 | DSC2017-38 | 1 | 1 | 0.0 |
| RM2004-14 | 7 | 55 | 1.7 | DSC2017-39 | 1 | 1 | 0.0 |
| RM2004-15 | 6 | 19 | 1.5 | RM2013-1 | 13 | 58 | 2.0 |

Multivariate analysis on the GTCrB and GTCrC ostracod assemblages was performed using the Xlstat software Addinsoft (2020) (XLSTAT statistical and data analysis solution, New York, USA; https://www.xlstat.com) except for the calculation of SIMPER and of diversity indices performed using PAST software (PAlaeontological STatistic, version 4.02) [64]. Cluster analysis was run for the samples (Q mode). The best results were reached using Ward's method and the Bray and Curtis algorithm.

*2.3. GIS Analysis*

Predictive distribution maps for critical species and the Shannon Index have been interpolated in GIS using the Inverse Distance weighting method (IDW). The measured relative frequency values surrounding the predicted location have been used to predict a value for any unsampled location in the study area based on the assumption that things that are close to one another are more alike than those that are farther apart. IDW is a weighted distance average and so the predicted value is limited to the range of the values used in the interpolation.

Unlike other interpolation methods—Such as Kriging—IDW does not make explicit assumptions about the statistical properties of the input data. IDW is often used when the input data do not meet the statistical assumptions of more advanced interpolation methods.

IDW assumes that each measured point has a local influence that diminishes with distance. It gives greater weight to points closest to the prediction location, and the weight decreases as a function of distance raised to a power value (p = 2 in our case). The search neighborhood can be altered by changing its size and shape and/or by changing the number of neighbors included. The maximum and minimum number of neighbor measures to include has been set, and the neighborhood search has been divided into sectors to account for any directional autocorrelation or trend in the data [65].

*2.4. Nutrient Analyses and Multiprobe Data Acquisition*

For nutrient analyses, surface water samples were collected with a horizontal Niskin bottle (V = 5 L). For determination of dissolved nutrients (ammonia, N—$NH_4^+$; nitrite, N—$NO_2^-$; nitrate, N—$NO_3^-$; soluble reactive silicate, SRSi; and soluble reactive phosphorus, SRP), samples were collected in HCl acid-washed polyethylene bottles (V = 100 mL) after filtration by means of GF/F fiber filters (Millipore, 0.45 μm) and immediately frozen (T = −20 °C) until analysis. The nutrients were always determined by means of the segmented flow technique (Bran + Luebbe AutoAnalyzer 3 and QuAttro) following the methods reported by Grasshoff [66] and modified for the specific instrument. The specific calculated method detection limits were 0.02 μM for N—$NH_4^+$, N—$NO_2^-$, N—$NO_3^-$, and 0.01 μM for SRP and SRSi. Certified standards (Inorganic Ventures Standard Solutions and MOOS-2, NRC) were used to ensure accuracy of the procedures. In addition, analytical performance was periodically checked through proficiency tests (PT) exercises organized by the European network of PT providers (QUASIMEME programmes AQ1 and AQ2).

During sampling, water column vertical profiles of pressure (dbar), temperature (°C), conductivity (mS/cm), salinity, pH, dissolved oxygen (% saturation and mg $L^{-1}$) (DO), and chlorophyll *a* (Chl *a*) (as an estimate of phytoplankton biomass, μg $L^{-1}$) from the surface to the bottom were collected on board using Idronaut mod. 316 (2004-05; 2013) and Idronaut mod. 316 plus (2014-17) multiparametric probes, which were calibrated following the manufacturer's protocols. The data obtained were processed using Idronaut software in order to verify the quality check.

All physicochemical data were processed in order to determine means, median, standard deviations, standard error, and maximum and minimum values and were graphically displayed as boxplots using the free PAST software version 2.06. Spearman correlation coefficients (r) indicate the strength and direction of a linear relationship between variables; r was considered significant when the *p*-value was <0.05. The trophic state was calculated by applying the trophic index TRIX [67]. This index combines nutrients (Dissolved Inorganic Nitrogen (DIN) as a sum of N–$NH_4^+$, N–$NO_2^-$, N–$NO_3^-$, and TP expressed as μg $L^{-1}$ of N and P, respectively), Chl *a* (μg $L^{-1}$), and DO (absolute deviation from % saturation).

## 3. Results

*3.1. Ostracod Evolution*

Fifty-two species were identified in the examined area: 37 and 24 species in GTCrB and GTCrC, respectively, while [48,60] found 25 species in the same area (Table S1 in the Supplementary Materials). Dominant species in the GTCrB samples are *Aurila convexa*, *Carinocythereis whitei*, *Cytheridea neapolitana*, *Leptocythere ramosa*, *Loxoconcha ovulata*, *Pseudopsammocythere similis*, *Pterygocythereis jonesii*, *Semicytherura incongruens*, while in GTCrC they are outnumbered by *A. convexa*, *L. ovulata*, *Loxoconcha rhomboidea*, *S. incongruens*, *Xestoleberis communis*, and *Xestoleberis dispar* (Figure A1).

In the same area, [48,60] recorded the prevailing presence of *Callistocythere adriatica*, *Callistocythere flavidofusca*, *Carinocythereis carinata*, *Cushmanidea elongata*, *C. neapolitana*, *Loxoconcha tumida*, *Palmoconcha turbida*, *P. jonesii*, *S. incongruens*, *X. communis*, and *X. dispar* (Table S1).

The H index ranges between 0 (samples DSC2017-38 and DSC2017-39) and 2.7 (sample TG2004-13). Higher values were recorded in samples from GTCrB with an evident drop in values in GTCrC (Table 1).

The comparison between the different analyzed periods showed continuity over time in the examined area of the following species: *A. convexa*, *C. adriatica*, *C. neapolitana*, *P. jonesii*, *S. incongruens*, *X. communis*, and *X. dispar*.

The species *Callistocythere flavidofusca*, *Cushmanidea elongata*, *Cytheretta subradiosa*, *Hiltermannicythere turbida*, *Leptocythere multipunctata*, *Loxoconcha avellana*, *Loxoconcha tumida*, and *Schedopontocypris setosa* were found only in GTCrA.

The Q-mode cluster analysis performed on ostracod associations found in GTCrB and GTCrC reveals the presence of three groups of samples (Figure 3).

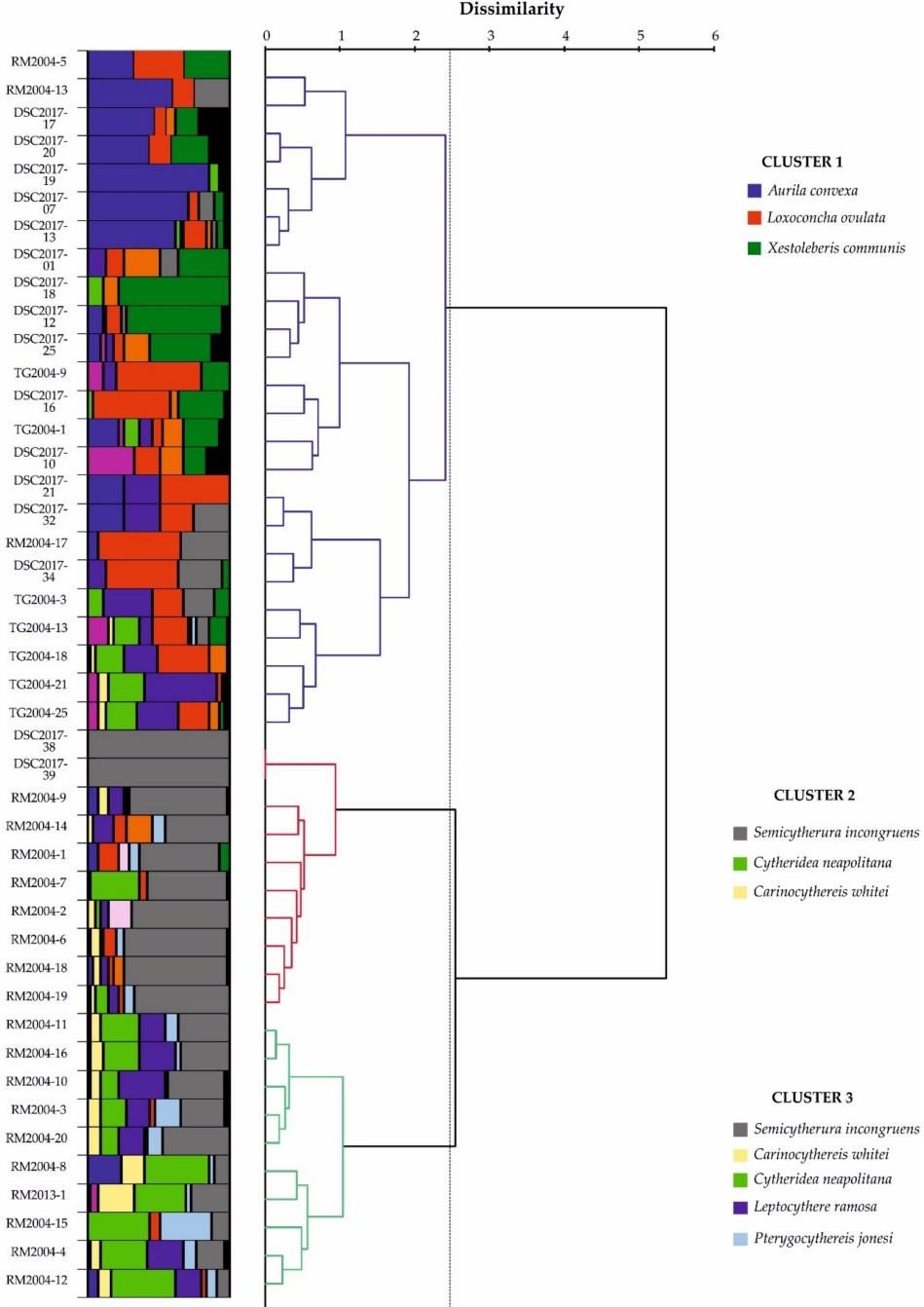

**Figure 3.** Q mode cluster analysis (Ward's method—Bray and Curtis algorithm): the dominant species for each cluster are reported. Authors' elaboration.

The SIMPER analysis shows that more than 60% of the difference between clusters is defined by *S. incongruens* (24.4% relative contribution), *A. convexa* (9.9%), *C. neapolitana* (9.9%), *L. ovulata* (8.6%), and *X. communis* (8.3%) followed by lower contributions of *Leptocythere ramosa* (6.8%), *P. jonesii* (3.8%), and *Carinocythereis whitei* (3.5%) (Table 2).

**Table 2.** Similarity Percentage (SIMPER) Analysis for Ostracod Assemblages Defined with Q-mode Cluster Analysis: The Dominant Species for Each Cluster are Reported in Bold. Overall Average Dissimilarity: 79.17.

| Species | Av. dissim | Contrib. % | Cumulative % | Cluster 1 | Cluster 2 | Cluster 3 |
|---|---|---|---|---|---|---|
| Semicytherura incongruens (Müller, 1894),Ruggieri, 1959 | 19.3 | 24.4 | 24.4 | 5.7 | **63.3** | **22.9** |
| Aurila convexa (Baird, 1850) | 7.8 | 9.9 | 34.3 | **18.2** | 1.7 | 3.2 |
| Cytheridea neapolitana Kollmann, 1958 | 7.8 | 9.9 | 44.2 | 4.3 | **4.3** | **25.4** |
| Loxoconcha ovulata (O.G. Costa, 1853) | 6.8 | 8.6 | 52.8 | **17.9** | **4.2** | 1.8 |
| Xestoleberis communis G. W. Müller, 1894 | 6.6 | 8.3 | 61.1 | **16.0** | 1.0 | 0.1 |
| Leptocythere ramosa (Rome, 1942) | 5.4 | 6.8 | 68.0 | 8.4 | 3.6 | **13.0** |
| Pterygocythereis jonesi (Baird, 1850) | 3.0 | 3.8 | 71.7 | 0.1 | 2.7 | **8.7** |
| Carinocythereis whitei (Baird, 1850) | 2.8 | 3.5 | 75.3 | 0.7 | 3.1 | **8.6** |
| Loxoconcha rhomboidea (Fischer, 1855) | 2.2 | 2.7 | 78.0 | 4.2 | 2.3 | 0.0 |
| Xestoleberis dispar G. W. Müller, 1894 | 1.9 | 2.3 | 80.3 | 4.3 | 0.4 | 1.1 |
| Pseudopsammocythere similis (Müller, 1894),Carbonnel, 1969 | 1.8 | 2.3 | 82.6 | 1.1 | 2.0 | 4.6 |
| Cistacythereis turbida (G. W. Müller, 1894) | 1.4 | 1.7 | 84.3 | 0.5 | 1.9 | 3.1 |
| Leptocythere bacescoi (Rome, 1942) | 1.2 | 1.5 | 85.8 | 0.0 | 3.2 | 1.2 |
| Cytheroma variabilis G. W. Müller, 1894 | 1.2 | 1.5 | 87.3 | 0.1 | 0.3 | 3.8 |
| Loxoconcha affinis (Brady, 1866) | 1.1 | 1.4 | 88.7 | 2.3 | 0.8 | 0.0 |
| Callistocythere adriatica Masoli, 1968 | 1.1 | 1.4 | 90.2 | 2.5 | 0.0 | 0.5 |

Cluster 1 is dominated by *A. convexa*, *L. ovulata*, and *X. communis*. Relatively low numbers of *C. neapolitana*, *Loxoconcha affinis*, and *L. rhomboidea* were recorded (specifically, within the samples DSC2017-01, RM2004-5, and TG2004-21). *S. incongruens* dominates cluster 2, followed by very low frequencies of *C. neapolitana* and *C. whitei* recorded in scattered samples. Although cluster 3 is still dominated by *S. incongruens*, it includes high concentrations of several other species, such as *C. neapolitana*, *C. whitei*, *L. ramosa*, and *P. jonesii* (Figure 3).

### 3.2. Physicochemical Variables

Table 3 lists the descriptive univariate statistic of the physicochemical variables considered in this study in the periods from 2004–2005 and 2013–2017. Water temperature (T) showed typical patterns of the Mediterranean area, with minimum values recorded at the beginning of March 2005 (5.99 °C) and maximum values in June 2013 at 28.2 °C. Salinity (S) values generally depend on the degree of freshwater inputs from the Isonzo River, which are strongly related to rainfall [68]. As a result, the early spring and autumn displayed the lowest values with an outlier of 9.0 recorded in November 2014, likely during a period of high discharge from the river. DO, expressed as % of saturation, ranged from 80.4 to 129%, whereas Chl *a*, which is a good estimate of phytoplankton biomass, ranged from 0.1 to 2.47 µg L$^{-1}$; thus, the occurrence of significant algal blooms can be excluded for both periods investigated. These results are comparable to those reported for the period from 1970 to 2007 in the whole Northern Adriatic basin [69].

N-NO$_3^-$ was the predominant form of DIN. In fact, on average, it accounted for 86.5% and 77.1% of total dissolved nitrogen for the periods from 2004–2005 and 2013–2017, respectively. The lowest mean values were found in summer (0.41 ± 0.45 µM, 2004–2005), whereas in spring and autumn, the N—NO$_3^-$ content increased due to riverine inputs. Certain European Directives give threshold values for DIN (74/440/EEC; 76/464/EC; 78/659/EC; 80/68/EC; 98/15/EC). Taking into consideration the whole data set, DIN did not exceed these values (DIN < 15 mg L$^{-1}$ N-1072 µM N; N—NO$_3^-$ < 25 mg L$^{-1}$ N-403 µM N; and N—NH$_4^+$ < 1 mg L$^{-1}$ N-71 µM N). Finally, SRP ranged from <loq to 0.36 µM P.

**Table 3.** Univariate Statistic for Physicochemical Parameters.

| 2004–2005 | T (°C) | S | Chl *a* μg L$^{-1}$ | O$_2$ (%) | N—NO$_2$ μM | N—NH$_4$ μM | N—NO$_3$ μM | P—PO$_4$ μM | Si—SiO$_2$ μM | TN μM | TP μM |
|---|---|---|---|---|---|---|---|---|---|---|---|
| N | 117 | 117 | 117 | 117 | 112 | 114 | 116 | 114 | 117 | 117 | 117 |
| Min | 5.99 | 28.7 | 0.10 | 82.9 | 0.01 | 0.01 | 0.08 | 0.01 | 0.15 | 5.11 | 0.17 |
| Max | 26.8 | 38.4 | 2.00 | 112 | 1.56 | 2.00 | 22.9 | 0.19 | 12.1 | 34.3 | 4.49 |
| Mean | 14.5 | 37.0 | 0.59 | 98.4 | 0.40 | 0.74 | 3.12 | 0.07 | 3.20 | 12.8 | 0.82 |
| Std. error | 0.54 | 0.15 | 0.04 | 0.57 | 0.04 | 0.04 | 0.36 | 0.00 | 0.21 | 0.52 | 0.06 |
| Stand. dev | 5.80 | 1.62 | 0.45 | 6.13 | 0.45 | 0.44 | 3.93 | 0.04 | 2.30 | 5.66 | 0.63 |
| Median | 13.6 | 37.4 | 0.50 | 98.1 | 0.21 | 0.71 | 1.90 | 0.06 | 2.60 | 11.2 | 0.66 |
| 25 prcntil | 9.24 | 36.8 | 0.20 | 93.7 | 0.07 | 0.36 | 0.74 | 0.04 | 1.48 | 9.18 | 0.53 |
| 75 prcntil | 18.5 | 37.9 | 0.80 | 104 | 0.53 | 1.04 | 4.29 | 0.09 | 4.42 | 14.6 | 0.93 |
| **2013–2017** | **T (°C)** | **S** | **Chl *a* μg L$^{-1}$** | **O$_2$ (%)** | **N—NO$_2$ μM** | **N—NH$_4$ μM** | **N—NO$_3$ μM** | **P—PO$_4$ μM** | **Si—SiO$_2$ μM** | **TN μM** | **TP μM** |
| N | 90 | 90 | 90 | 89 | 88 | 90 | 90 | 87 | 86 | 88 | 86 |
| Min | 7.89 | 9.00 | 0.10 | 80.4 | 0.02 | 0.02 | 0.02 | 0.01 | 0.18 | 1.84 | 0.01 |
| Max | 28.2 | 38.3 | 2.47 | 129 | 2.83 | 13.9 | 59.0 | 0.36 | 83.3 | 89.1 | 3.26 |
| Mean | 17.2 | 34.6 | 0.70 | 101 | 0.38 | 1.75 | 6.88 | 0.06 | 8.38 | 18.1 | 0.14 |
| Std. error | 0.60 | 0.44 | 0.05 | 0.92 | 0.06 | 0.22 | 1.09 | 0.01 | 1.16 | 1.77 | 0.04 |
| Stand. dev | 5.71 | 4.16 | 0.43 | 8.69 | 0.56 | 2.11 | 10.4 | 0.07 | 10.8 | 16.6 | 0.35 |
| Median | 16.9 | 36.1 | 0.68 | 101 | 0.19 | 0.99 | 4.35 | 0.03 | 5.97 | 12.1 | 0.08 |
| 25 prcntil | 12.3 | 33.1 | 0.35 | 95.5 | 0.05 | 0.47 | 1.88 | 0.01 | 2.69 | 9.50 | 0.06 |
| 75 prcntil | 22.2 | 37.2 | 0.83 | 107 | 0.41 | 2.16 | 6.68 | 0.07 | 10.1 | 19.6 | 0.12 |

The DIN–SRP molar ratio is commonly used to detect whether N and P act as factors capable of limiting primary production [70]. In this work, the ratio was always higher than 16, suggesting that the system is P-limited [71].

Several criteria are commonly used to define the trophic state in aquatic systems. The TRIX index was set by [72], and according to these authors, the quality varied from high, characteristic of a system with low productivity and low trophic level (TRIX: 2–4), to poor, typical of a highly productive system with high trophic levels (TRIX: 6–8). In this work, TRIX ranged from 2.19 to 4.06 (average based on seasonal aggregated data), which is consistent with a low trophic level and good water quality, especially during summer periods.

Pearson linear correlations between variables are shown in Figure 4, where the significant correlations ($p < 0.05$) are boxed. A strong negative correlation of S with oxidized nitrogen (N—NO$_3$), SRSi, TP, and TN was observed, especially during the period from 2013–2017, whereas Chl *a* contents were scarcely correlated with nutrients, thus indicating that nutrient inputs are not sufficient to cause significant primary productivity. Nutrients are positively correlated amongst themselves, suggesting their common origin.

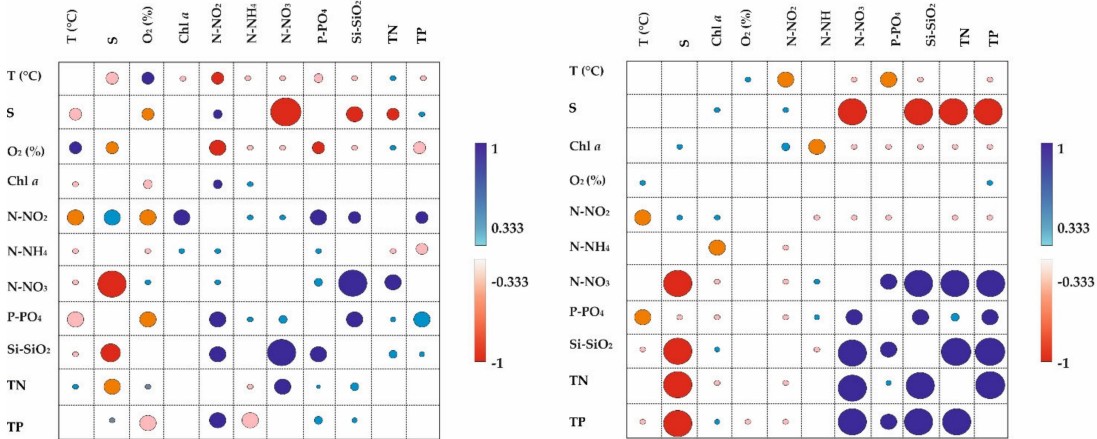

**Figure 4.** Pearson linear correlations between physicochemical variables. Significant correlations ($p < 0.05$) (positive in blue and negative in red) are boxed. Color intensity and the size of the circle are proportional to the correlation coefficients. Authors' elaboration.

The results obtained for some physicochemical parameters (T, S, Chl *a*, N—$NO_3^-$, and SRP) in the two periods were compared in order to check if a significant increase or decrease had occurred: these factors can influence the distribution, richness, and diversity of benthic organisms. For these purposes, we applied the Kruskal–Wallis test for equal medians. The results showed a significant increase in T, Chl *a*, and N-$NO_3^-$, whereas S and SRP significantly decreased.

## 4. Discussion

Recent numerous works have highlighted the variations and impact on ecosystems of recent growing urbanization at a global level. Urbanization may filter out species that are not preadapted to urban conditions, with a subsequent decrease in abundance or diversity at the small (local) scale [8,73].

Alternatively, the loss of species less adapted to urban environments could be (over)compensated by an increase in species efficient in exploiting urban resources [2,3,74]. Both phenomena may cause biotic homogenization if local communities are colonized by the same species, in turn increasing the compositional similarity of urban species assemblages and, consequently, reducing species richness of urban areas on a large scale [2,75]. The relationship between the growth of cities and the impact of human activities on adjacent marine areas is still currently difficult to analyze. Yasuhara et al. [5] and Wilkinson et al. [45] recorded how, in marine realms, the predominant cause of degradation noticed in microfossil records was nutrient enrichment and the resulting symptoms of eutrophication including hypoxia.

In particular, in the Adriatic Sea, changes were caused by eutrophication and anoxia due to human activities including agriculture, wastewater disposal, and diversion of river outflow [12,13]. In this sense, ostracods are usually intolerant to hypoxia and respond with a reduction in diversity and richness, and in some cases, populations become monospecific [46,47].

In the northern Adriatic ecosystems and the GoT area, a review of numerous long-term studies on river discharges, oceanographic features, plankton, fish, and benthic compartments collected since the 1970's revealed significant changes in mechanisms and trophic structures [55,76]. In detail, a gradual increase in eutrophication phenomena, characterized by significant hypoxic events at the bottom, was recorded during the 1970s until the mid-1980's [77–79], followed by a reversal of the trend, particularly marked in the 2000s [69,80]. This trend was attributed to a combination of the reduction in anthropogenic impact, mainly due to a substantial decrease of the phosphorus loads and of climatic modifications, resulting in decline in atmospheric precipitations and, consequently, of runoff [81,82].

The long-term data set (1986–2010) of phytoplankton abundance, used to investigate the temporal variability of the phytoplankton community at a coastal site in the GoT, appears to confirm previous analysis. The interannual variability of the phytoplankton community shows two major periods in terms of abundance and community composition. The first one, 1986–1994, characterized by the highest abundances of microalgae and the dominance of phytoflagellates and the second period, 1995–2007, showed lower abundances and a collapse of the phytoplankton. Lastly, an apparent new increase in abundances has been recorded in recent years (2008–2010). The observed long-term changes could be related to the more general oligotrophication of the northern Adriatic Sea [80,83].

In particular, oligotrophication in the GoT can be ascribed to a reduction in outflow from the Isonzo River observed during the period from 1986 to 2010, with occurrence of dry years in the latter part of the period in 2003, 2005, 2006, and 2007 [81,84].

However, occurrence of significant atmospheric phenomena due to climate change is indicated by the increase in N-$NO_3^-$ and the decrease in salinity: long drought periods are followed by heavy rainfalls that increase Isonzo River discharge for short periods [81]. Concomitant with dry periods, the phytoplankton community time series showed the absence of some spring (2005 and 2007) and autumn (2005, 2006, and 2007) blooms, highlighting the possible direct relationship between the external input and the productivity of the ecosystems. [80,83].

Analysis of the molluscan community composition in the Bay of Panzano cores [4,28] recorded how frequent past hypoxic events intensifying pressure from fishing and climatic factors can replace

contamination as the main drivers of community change, leading to the most pronounced shifts in molluscan community composition. In the second half of the 20th century, disturbances from fishing and hypoxia intensified with the benefit of opportunistic species. In particular, the shift in abundance and the increase in the size of the opportunistic species *Corbula gibba* recorded in the Bay of Panzano and the loss of formerly abundant, hypoxia-sensitive species in and around 1950 coincided with the higher preservation of organic matter and higher frequency of seasonal hypoxia [85,86].

Results from the foraminiferal record in the same cores confirm eutrophication as the most significant driver of community shifts [26].

Finally, direct biological observations showed that seasonal mass mortalities in the Adriatic Sea in the late twentieth century negatively affected predators and substrate-destabilizing bioturbators, including burrowing shrimps, infaunal echinoids, holothurians, predatory asteroids, and muricid gastropods. The recovery of these taxa in the wake of hypoxic events in the northern Adriatic Sea is delayed and occurs over several years [87].

To verify the effects of urbanization on ostracod fauna over the last 50 years, the experimental site was carefully chosen at the center of the innermost part of the GoT. Moreover, presence within the area of the MPA (EUAP 0167), established in 1986 by a decree from the Italian Ministry of the Environment, which has entrusted its management to the WWF Italy onlus Association (D.M. November 12, 1986) and, from 2013, was included in the SCI (Site of Community Importance) list (directives 79/409/EEC and 92/43/EEC) (http://www.riservamarinamiramare.it), afforded the opportunity to analyze potential ostracod response in environmental subjects to varying degrees of environmental stress.

Sediments in the GoT are also dispersed regularly in concentric bands with respect to the mouth of the main rivers with dominant sedimentation in the experimental site, mainly characterized by sandy pelite and very sandy pelite [51]. All samples examined are additionally included at a reduced depth range so both these parameters can be considered irrelevant in the distribution of ostracod association.

Analysis of ostracod fauna showed a clear quantitative/qualitative decrease from GTCrB to GTCrC. The only exception is sample RM2013-1 collected in the MPA, which shows values comparable with GTCrB associations (Table S1). The Shannon index values further confirm the above data with a clear decrease in all recent samples except for RM2013-1 (Figure 5).

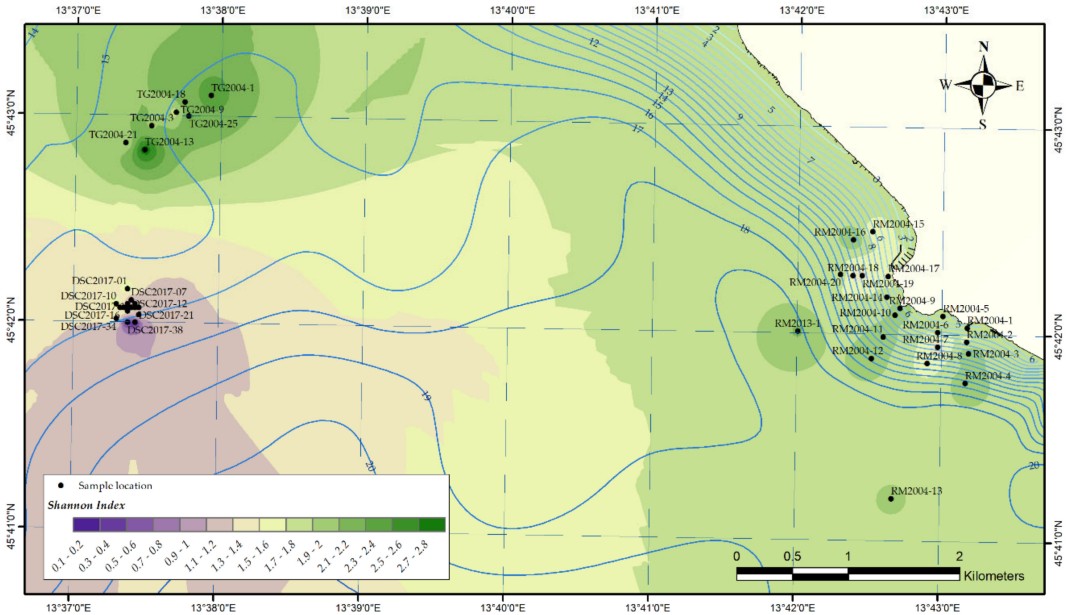

**Figure 5.** Shannon Index interpolated in GIS using the Inverse Distance weighting method (IDW): Authors' elaboration.

The indications coming from the analyzed species over a time span of 20 years (GTCrB and GTCrC) records in both samplings a dominance of species tolerant to high organic matter concentrations and oxygen deficiency such as *C. neapolitana*, *L. ramosa*, and *Loxoconcha* spp. [88].

In addition, in GTCrB, the dominant fauna is represented by the species *S. inconguens*, *C. neapolitana*, *L. ramosa*, *P. jonesi*, and *C. whitei*, and in GTCrC samples, the trend towards homogenization is further amplified with the domain of few different opportunistic ostracod faunas represented by *A. convexa*, *Loxoconcha* spp., and *Xestoleberis* spp. [5,33,45,89] (Figure 6A,B). In addition, Salvi et al. [43] recorded the presence of living specimens of *X. communis* and *X. dispar* in stressful environmental conditions such as the Ex-Military Arsenal of the La Maddalena Harbor; both species were found to be dominant at GTCrC.

In GTCrA, even when cautiously viewed through the lens of qualitative analysis, the ostracod association from 1967 differs from those found recently. The number of species reported by Masoli [48,60] is numerically lower than GTCrB, but the association shows the presence of species no longer found in recent samples (*C. flavidofusca*, *C. elongata*, *C. subradiosa*, *H. turbida*, *L. multipunctata*, *L. avellana*, *L. tumida*, and *S. setosa*) (Table S1). Among these, *C. subradiosa* and *Loxoconcha* spp. are also recognized in the literature as among the most tolerant to environmentally stressful conditions [35,90].

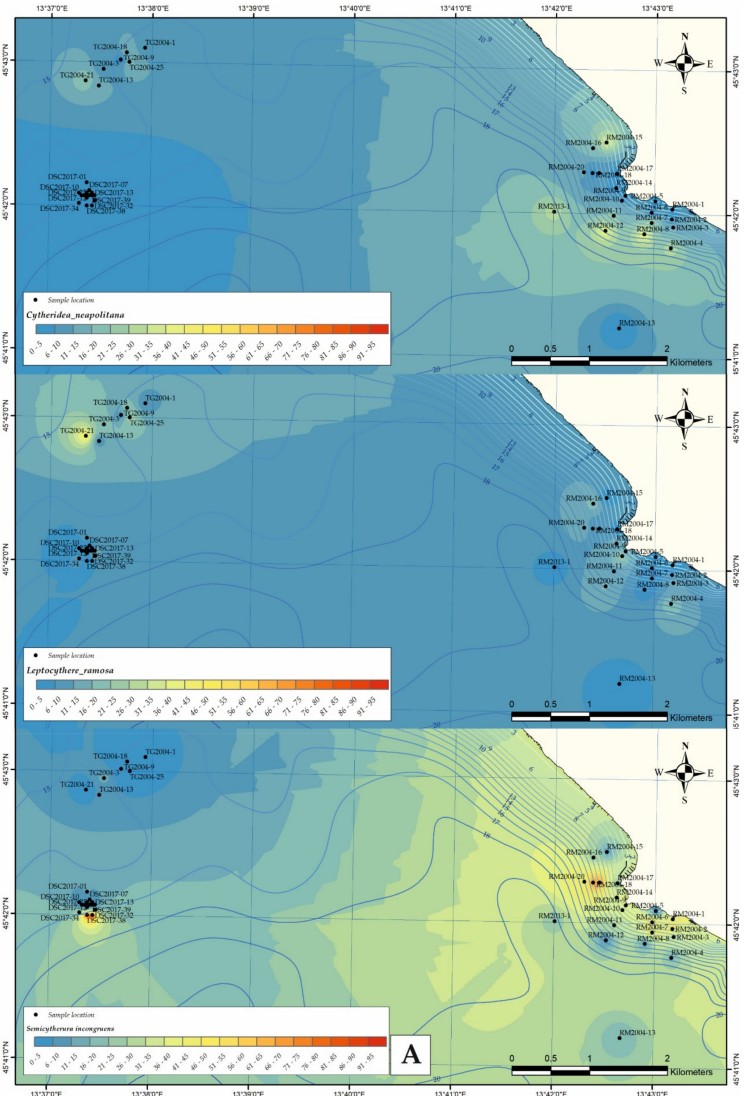

**Figure 6.** *Cont.*

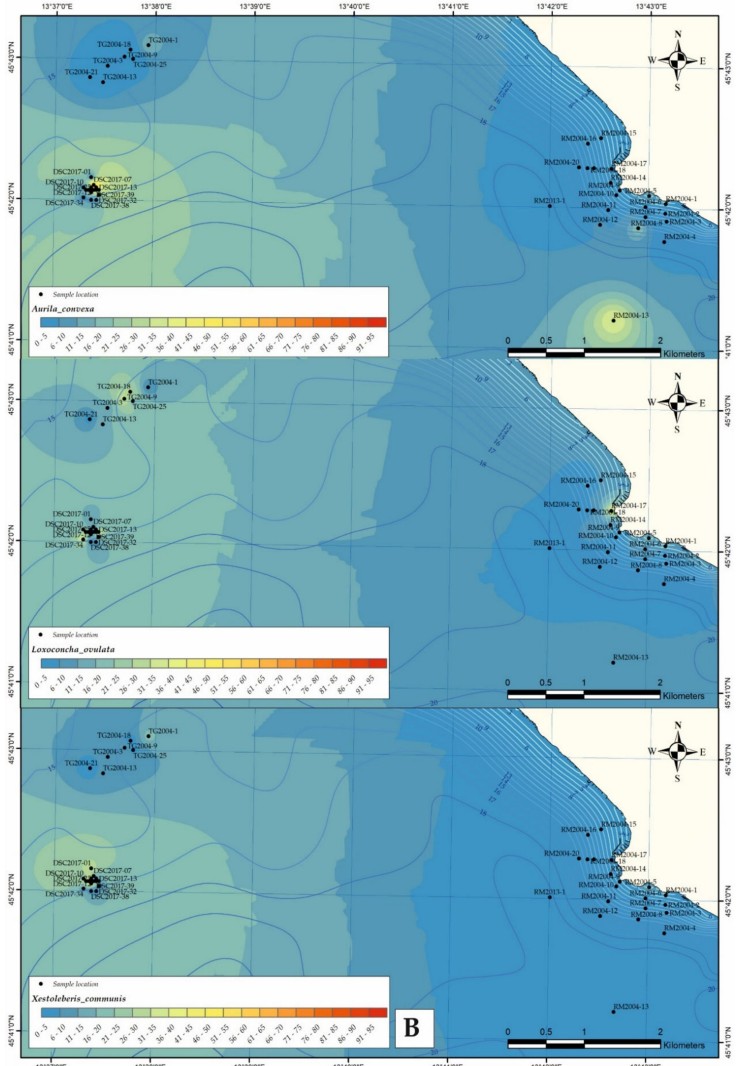

**Figure 6.** Diffusion of the main opportunistic species in GTCrB (2004–2005) (*C. neapolitana, L. ramosa, and S. incongruens*) (**A**) and GTCrC (2013–2017) (*A. convexa, L. ovulata,* and *X. communis*) (**B**), interpolated in GIS using the Inverse Distance Weighting Method (IDW): Authors' elaboration.

Considering the length of time which passed between the samples being taken (1967, 2004–2017) and in the absence of intermediate sampling activities to indicate with certainty the precise evolution of ostracod associations over the last 50 years, it is at any rate possible to summarize the response of ostracod associations to the anthropic evolution of the area:

(1) Since the nineteen-sixties, ostracods have been exposed to a potentially compromised environment due to the possible increase in anoxia phenomena [4,91], with few prevailing species tolerant to environmentally stressful conditions. The data from the ostracod associations recorded by Masoli [48,60] show a compromised environmental situation with few dominant species including the opportunistic species *C. subradiosa, Loxoconcha* spp., and *Xestoleberis* spp. present in almost all samples analyzed;

(2) Analysis of ostracod assemblages indicate a slight recovery in qualitative and quantitative terms in the early 2000s, with the majority of species found being adapted to high nutrient supply and tolerant to hypoxia. However, the ostracod association confirms environmental stress caused by frequent crises linked to the recurrence of mucilage and hypoxic events documented in particular in the second half of the 20th century [77–79] as highlighted by the studies of molluscan and foraminiferal associations in the Bay of Panzano [4,28,86]. The small number of ostracod

species found in GTCrB (*S. inconguens*, *C. neapolitana*, *L. ramosa*, *P. jonesi*, and *C. whitei*) and their characteristics of opportunism and resistance to environmental stress testify to an environment subject to a strong meiofauna depletion (Figure 6A). In fact, ostracod abundance recovers slowly after the cessation of hypoxia, unlike that of other benthos. One reason for this low level of year-round abundance may be that ostracods do not have a planktonic juvenile stage; therefore, their ability to migrate is very low [41,92]. Analyses of the ostracod associations found in the Bay of Panzano confirm the abovementioned environmental crises with a clear qualitative/quantitative decrease over the last 50 years (N. Pugliese personal communication, June 2020);

(3)  Recent ecological studies reported significant modifications of the environmental conditions in the GoT due to climatic fluctuations and changes in anthropogenic pressure as follows: 1—a warming of surface waters at the regional scale [50,55]; 2—a marked decrease in the freshwater outflow during the 2000s due to a reduction in precipitation [93] and an increased river load of N coupled with decreased P loads due to enforcement of environmental law [10,81]; 3—an increased DIN/$PO_4$ ratio in the waters due to the reduction of riverine TP and to a limitation of N uptake by phytoplankton [94,95]. These changes have led to a situation of marked oligotrophy where the reduced nutrient content (detergent phosphate limitations; improvement of the sewage filtration systems in the urban area of Trieste in response to the community infringement procedure to which the Friuli Venezia Giulia Region has been subjected since 2004) linked to ongoing climate change phenomena could have induced a further, more serious crisis in ostracod associations after the anoxia crises of previous decades (Table A1).

In fact, from GTCrB to GTCrC, an evident decrease in ostracods is recorded together with modification of ostracod associations despite the presence of a marine protected area in the experimental site since 1986 (Figure 6A,B).

The prevailing presence of the species *A. Convexa*, *Loxoconcha* spp., and *Xestoleberis* spp. in GTCrC samples, all characterized by a high degree of opportunism and resistance to environmental stress, could confirm the recent rise in oligotrophic phenomena (Figure 6B, Table A1).

The rapid disappearance of Phanerogams, accelerated in 2015, and the strong decrease in the phytoplankton community in the late winter-early spring bloom observed in recent years (2010–2017) [80,83], from the GoT in general and in the experimental site in particular, could represent further evidence of the changed environmental conditions linked to increasing anthropic stress and/or climatic change.

Finally, it must be underlined how the best environmental conditions in recent samples have been recorded in the MPA. This is not entirely surprising since meiofauna are commonly early colonists and that the most mobile and sensitive taxon rapidly colonize sediments where favorable conditions are restored [96].

## 5. Conclusions

Analysis of ostracod associations found in the GoT during the period from 2004–2017 showed how, over the last 20 years, there has been a decline in environmental conditions with a clear decrease in qualitative/quantitative values. In particular, in GTCrB, a higher number of species was observed than in GTCrC, with a clear decrease in living specimens, but in both cases, we highlighted a trend towards homogenization in the examined area with the domain of few species in most samples, and in some cases, samples were found to be monospecific. Most of the species recovered have characteristics of opportunism and tolerance to environmentally stressful conditions. *A. convexa*, *C. neapolitana*, *Loxoconcha* spp., *S. incongruens*, and *Xestoleberis* spp. are all known to be hypoxia-tolerant or opportunistic and are able to survive in areas of severe environmental conditions such as the polluted Ex-Military Arsenal of La Maddalena Harbor.

The study on ostracod assemblages found in GTCrA, even when exercising caution with regards to a purely qualitative analysis, also records environmentally stressful conditions with the presence of few species often recognized in the literature as among the most tolerant to poor environmental conditions.

These data are in agreement with the analysis performed in the GoT on other taxa (Mollusks and Foraminifera), thus confirming the possible environmental crisis linked to recurrence of mucilage and hypoxic events, documented for the GoT between the 1980s and the first decade of the 21st century.

The decrease in ostracods in GTCrC might be related to the rapid disappearance of Phanerogams, accelerated in 2015, and to the marked decrease in the phytoplankton community in recent years (2010–2017) due to an improvement of the oligotrophic conditions, markers of the changed environmental conditions linked to increasing anthropic stress and/or climatic change.

In addition, our work may extend the knowledge about sensitive and more tolerant species (surviving post-disturbance "winners") and can help pinpoint and define the spatial extension of past, present, and potential future mortalities and/or disappearance linked to growing anthropization. In the same way, the roles individual species play help better gauge potential effects on ecosystem integrity, function, and resilience.

The best environmental conditions in recent samples have been recorded in the MPA, therefore indicating that the preservation of large and connected patches of natural habitats is the most effective measure to halt further urbanization-driven biodiversity loss.

Future study on cores collected in selected areas of the GoT will increase the understanding of the repercussions of anthropogenic activities over time on ostracod assemblages as well as will identify additional indicator species, through seasonal sampling, that can be used to better define (a) possible causes of the recent decline in ostracod associations, (b) the status and vulnerability of the ecosystem, and (c) evaluation of remediation activities to mitigate the negative impact of urbanization.

**Supplementary Materials:** The following are available online at http://www.mdpi.com/2071-1050/12/17/6954/s1, Table S1: List of recognized living ostracod species (%) recovered in GoT during 2004–2017. x indicates the species found by Masoli, 1967.

**Author Contributions:** The paper is the result of the shared reflections, research and work mainly of the authors G.S. and A.A. However, Abstract and Conclusions are shared by G.S., A.A., N.P. and S.C. (Saul Ciriaco). Section 1 is attributed to G.S. and A.A.; In Section 2, Sections 2.1 and 2.2 is to be attributed to G.S., Section 2.3 to M.F. and Section 2.4 to A.A.; In Section 3, Section 3.1 is to be attributed to G.S. and Section 3.2 to A.A; Section 4 was written by G.S.; Cartographic Maps and figures elaborations have been drawn up by G.S. and M.F. (Figures 1–5 and 6A,B); Tables 1 and 2, were edited by G.S.; Table 3 was edited by A.A.; M.C. was responsible for the water samples collection and the processing of chemical-physical data; S.C. (Stefano Cirilli) leaded for the sediment samples collection in 2017 marine survey. G.S. realized Figure A1 and Table A1, edited data in Appendix A. Supplementary Materials Table S1 was edited by G.S. All authors have read and agreed to the published version of the manuscript.

**Funding:** This research received no external funding.

**Acknowledgments:** We would like to acknowledge the reviewer comments suggested. Their input allowed us to further improve our manuscript. Special thanks to Karry Close for proofreading the manuscript.

**Conflicts of Interest:** Authors declare no conflict of interest.

## Appendix A

**Table A1.** Univariate Statistic for Dominant Ostracod Species (%) and Physicochemical Parameters (2004–2005 and 2013–2017 years): The Increasing Values from GTCrB to GTCrC of the Dominant Ostracod Fauna (Red Color) and the Corresponding Declining of the Nutrient Values (Phosphorus and Phosphates) (Blue Color) are Reported.

| 2004–2005 | N | Min | Max | Mean | Stand. dev | Median | 25 prcntil | 75 prcntil |
|---|---|---|---|---|---|---|---|---|
| *A.convexa (%)* | 27 | 0.00 | 60.00 | 6.86 | 13.98 | 1.19 | 0.00 | 7.14 |
| *C.whitei (%)* | 27 | 0.00 | 17.14 | 5.23 | 4.37 | 6.06 | 0.00 | 8.47 |
| *C.neapolitana (%)* | 27 | 0.00 | 45.71 | 16.27 | 15.57 | 13.58 | 0.00 | 27.16 |
| *L.ramosa (%)* | 27 | 0.00 | 58.54 | 14.54 | 14.58 | 11.11 | 0.00 | 25.42 |
| *L.ovulata (%)* | 27 | 0.00 | 66.67 | 13.97 | 17.92 | 5.00 | 2.17 | 22.22 |
| *P.jonesi (%)* | 27 | 0.00 | 37.50 | 5.05 | 8.07 | 2.44 | 0.00 | 8.57 |
| *S.incongruens (%)* | 27 | 0.00 | 83.33 | 32.89 | 27.47 | 32.31 | 8.70 | 57.50 |
| *X.communis (%)* | 27 | 0.00 | 35.00 | 5.16 | 9.96 | 0.00 | 0.00 | 5.13 |
| T (°C) | 117 | 5.99 | 26.8 | 14.5 | 5.80 | 13.6 | 9.24 | 18.5 |
| S | 117 | 28.7 | 38.4 | 37.0 | 1.62 | 37.4 | 36.8 | 37.9 |

**Table A1.** *Cont.*

| 2004–2005 | N | Min | Max | Mean | Stand. dev | Median | 25 prcntil | 75 prcntil |
|---|---|---|---|---|---|---|---|---|
| Chl *a* mg L$^{-1}$ | 117 | 0.10 | 2.00 | 0.59 | 0.45 | 0.50 | 0.20 | 0.80 |
| O$_2$ (%) | 117 | 82.9 | 112 | 98.4 | 6.13 | 98.1 | 93.7 | 104 |
| N-NO$_2$ µM | 112 | 0.01 | 1.56 | 0.40 | 0.45 | 0.21 | 0.07 | 0.53 |
| N-NH$_4$ µM | 114 | 0.01 | 2.00 | 0.74 | 0.44 | 0.71 | 0.36 | 1.04 |
| N-NO$_3$ µM | 116 | 0.08 | 22.9 | 3.12 | 3.93 | 1.90 | 0.74 | 4.29 |
| P-PO$_4$ µM | 114 | 0.01 | 0.19 | 0.07 | 0.04 | 0.06 | 0.04 | 0.09 |
| Si-SiO$_2$ µM | 117 | 0.15 | 12.1 | 3.20 | 2.30 | 2.60 | 1.48 | 4.42 |
| TN µM | 117 | 5.11 | 34.3 | 12.8 | 5.66 | 11.2 | 9.18 | 14.6 |
| TP µM | 117 | 0.17 | 4.49 | 0.82 | 0.63 | 0.66 | 0.53 | 0.93 |
| **2013–2017** | **N** | **Min** | **Max** | **Mean** | **Stand. dev** | **Median** | **25 prcntil** | **75 prcntil** |
| *A.convexa (%)* | 17 | 0.00 | 92.31 | 25.04 | 31.80 | 11.11 | 0.00 | 57.76 |
| *C.whitei (%)* | 17 | 0.00 | 26.19 | 1.54 | 6.35 | 0.00 | 0.00 | 0.00 |
| *C.neapolitana (%)* | 17 | 0.00 | 38.10 | 3.90 | 9.49 | 0.00 | 0.00 | 4.00 |
| *L.ramosa (%)* | 17 | 0.00 | 25.00 | 5.20 | 8.91 | 0.00 | 0.00 | 9.58 |
| *L.ovulata (%)* | 17 | 0.00 | 59.09 | 19.40 | 20.31 | 13.33 | 0.00 | 37.50 |
| *P.jonesi (%)* | 17 | 0.00 | 4.76 | 0.28 | 1.15 | 0.00 | 0.00 | 0.00 |
| *S.incongruens (%)* | 17 | 0.00 | 100.00 | 18.75 | 32.44 | 3.45 | 0.00 | 26.78 |
| *X.communis (%)* | 17 | 0.00 | 87.50 | 25.83 | 29.62 | 7.69 | 0.00 | 50.00 |
| T (°C) | 90 | 7.89 | 28.2 | 17.2 | 5.71 | 16.9 | 12.3 | 22.2 |
| S | 90 | 9.00 | 38.3 | 34.6 | 4.16 | 36.1 | 33.1 | 37.2 |
| Chl *a* mg L$^{-1}$ | 90 | 0.10 | 2.47 | 0.70 | 0.43 | 0.68 | 0.35 | 0.83 |
| O$_2$ (%) | 89 | 80.4 | 129 | 101 | 8.69 | 101 | 95.5 | 107 |
| N-NO$_2$ µM | 88 | 0.02 | 2.83 | 0.38 | 0.56 | 0.19 | 0.05 | 0.41 |
| N-NH$_4$ µM | 90 | 0.02 | 13.9 | 1.75 | 2.11 | 0.99 | 0.47 | 2.16 |
| N-NO$_3$ µM | 90 | 0.02 | 59.0 | 6.88 | 10.4 | 4.35 | 1.88 | 6.68 |
| P-PO$_4$ µM | 87 | 0.01 | 0.36 | 0.06 | 0.07 | 0.03 | 0.01 | 0.07 |
| Si-SiO$_2$ µM | 86 | 0.18 | 83.3 | 8.38 | 10.8 | 5.97 | 2.69 | 10.1 |
| TN µM | 88 | 1.84 | 89.1 | 18.1 | 16.6 | 12.1 | 9.50 | 19.6 |
| TP µM | 86 | 0.01 | 3.26 | 0.14 | 0.35 | 0.08 | 0.06 | 0.12 |

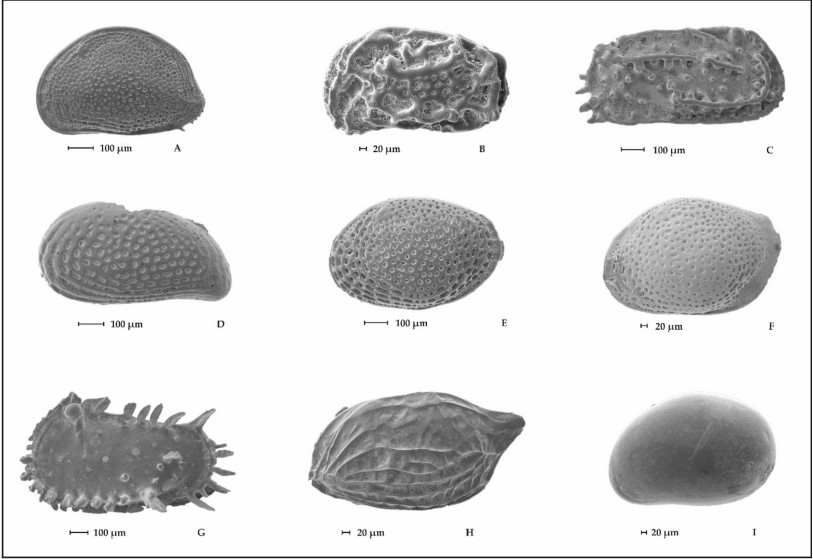

A.  *Aurila convexa*. Left valve, lateral exterior view. Scale bar = 100 µm;
B.  *Callistocythere adriatica*. Left valve, lateral exterior view. Scale bar = 20 µm;
C.  *Carinocythereis whitei*. Right valve, lateral exterior view. Scale bar = 100 µm;
D.  *Cytheridea neapolitana*. Right valve, lateral exterior view. Scale bar = 100 µm;
E.  *Loxoconcha ovulata*. Left valve, lateral exterior view. Scale bar = 100 µm;
F.  *Loxoconcha rhomboidea*. Left valve, lateral exterior view. Scale bar = 100 µm;
G.  *Pterygocythereis jonesi*. Left valve, lateral exterior view. Scale bar = 100 µm;
H.  *Semicytherura incongruens*. Left valve, lateral exterior view. Scale bar = 20 µm;
I.  *Xestoleberis communis*. Left valve, lateral exterior view, sample UC09. Scale bar = 20 µm.

**Figure A1.** SEM photomicrographs of the GoT dominant ostracod taxa.

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
