# Peer review of "Ostracod Fauna: Eyewitness to Fifty Years of Anthropic Impact in the Gulf of Trieste. A Potential Key to the Future Evolution of Urban Ecosystems"

_sustainability, doi:10.3390/su12176954_

Round 1
Reviewer 1 Report
The article is interesting, certainly needs to be published after additions and corrections.
The values of Stand. Dev need to be checked in Table 3: either they are too high, or 25 prcntil and 75 prcntil are calculated incorrectly.
The main result of the researches is shown graphically at the bottom of Figure 2: I recommend making this part of the figure more readable.
In table 2 you need to indicate the author who described each species and year of description. I also consider it useful to give the characteristics of the taxonomic belonging of species at the level above the generic one: either in Table 2, or in the lower part of Figure 2, or both.
In Table 3 the rounding off of the values is not in accordance with publishing standards. The same applies to the dimension in the column headings in this table.
Figure 4 does not have a sufficiently uniform distribution of points, so the predicted Shannon Index values on it interpolated are not correct. I recommend deleting it. The same goes for Figure 5: one point cannot change the picture on the map.
In Figure 5 the font is unreadable.
The results of the physico-chemical variables in the manuscript are weakly related to the species composition of ostracods. For such a large initial data set, more interesting statistical processing of the results can be carried out. The title of the article "Ostracod Fauna: ... Future Evolution of Urban Ecosystems" also implies an analysis of the distribution of certain species (at least dominant ones) in the gradients of factors shown in Figure 3. Analysis of the tolerance ranges of individual species in the studied water area (either in the form of a table with means, median, standard deviations, 25 and 75 prcntil, maximum and minimum values, or graphically with Box-analysis) will allow predicting further changes in the ostracod community with increasing impact Urban Ecosystems to the marine environment in other parts of their range, not just the surveyed area.
Author Response
Authors: Kind reviewer thank you for the useful suggestions and corrections. Here is our comments and answers to your report. We also point out that the new parts in the text are inserted in yellow color.
The values of Stand. Dev need to be checked in Table 3: either they are too high, or 25 prcntil and 75 prcntil are calculated incorrectly.
Author: Table 3 has been carefully checked and all the statistical parameters have been re-calculated by using PAST statistical software. The standard error has been added in the revised Table 3.
In Table 3 the rounding off of the values is not in accordance with publishing standards. The same applies to the dimension in the column headings in this table.
Author: The values have been corrected and the related dimensions were added.
The main result of the researches is shown graphically at the bottom of Figure 2: I recommend making this part of the figure more readable.
Authors: Figure 2 now 3 has been improved by making it more readable
In table 2 you need to indicate the author who described each species and year of description. I also consider it useful to give the characteristics of the taxonomic belonging of species at the level above the generic one: either in Table 2, or in the lower part of Figure 2, or both.
Authors: The authors of the species have been added in table 2. The same data are indicated in table S1 which, given the size, has been placed in supplementary materials in order not to make the text heavier.
With regard to the ostracod studies, it is not used in publications to refer to the level above the generic one with the exception of work of a purely taxonomic nature;
Figure 4 does not have a sufficiently uniform distribution of points, so the predicted Shannon Index values on it interpolated are not correct. I recommend deleting it. The same goes for Figure 5: one point cannot change the picture on the map.
We know that the best results with IDW could be obtained when sampling is dense and not so uneven like the set of samples we are presenting in this paper (three quite distant locations with many closer points) but this interpolation method is the more straightforward if compared to more advanced interpolation methods that require even more statistical assumptions. We compensate undesirable effects due to the distribution by including a large radius/number of neighbor points to estimate the unknown value, by dividing the neighborhood in sectors and keeping a low power factor of 2.
In Figure 5 the font is unreadable.
Authors: Figure 5 now 6 has been modified and has been made more legible
The results of the physico-chemical variables in the manuscript are weakly related to the species composition of ostracods. For such a large initial data set, more interesting statistical processing of the results can be carried out. The title of the article "Ostracod Fauna: ... Future Evolution of Urban Ecosystems" also implies an analysis of the distribution of certain species (at least dominant ones) in the gradients of factors shown in Figure 3. Analysis of the tolerance ranges of individual species in the studied water area (either in the form of a table with means, median, standard deviations, 25 and 75 prcntil, maximum and minimum values, or graphically with Box-analysis) will allow predicting further changes in the ostracod community with increasing impact Urban Ecosystems to the marine environment in other parts of their range, not just the surveyed area.
Authors: Indeed, we agree with the observation on the poor relationship between the chemical-physical parameters and the dominant species in the area during the period considered. We performed a statistical correlation matrix analysis but the results obtained are not statistically so significant.
The opportunistic meiofauna, as explained in the work, moves rapidly in search of favorable environments. In the case of the Gulf of Trieste the environmental situation is recently compromised by a strong oligotrophic regime as we have tried to highlight in our work by broadening, in the discussions, the possible relationship between the ostracods diffusion and human impact.
We have, however, included in the appendix a table with means, median, standard deviations, 25 and 75 prcntil, maximum and minimum values of dominant species in the different periods considered.
Authors: The text has been revised and corrected by the English language expert of the Department of Mathematics and Geosciences dr. Karry Close
Reviewer 2 Report
In the manuscript “Ostracod Fauna: Eyewitness to Fifty Years of Anthropic Impact in the Gulf of Trieste. A Potential Key to the Future Evolution of Urban Ecosystems” the Authors reported an interesting study on the long-term impact of anthropic activities on ostracods in the Gulf of Trieste. They highlighted the need of conservation strategies and the important role of protected areas. The paper is well-written and is suitable for Sustainability.
I only suggested some minor revisions:
Line 25: “…Foraminifera, have been impaired by the possible…”
Line 27: “GoT”
Line 48: “…Plain. The presence…”
Line 59: “Gallmetzer et al. [4] recorded…”
Line 65: “Ostracods are small… in length) which…”
Line 83: “…published by Masoli [46]…”
Line 93: Is 25 or 10 m the average depth?
Line 124: “…reported by Masoli [46]…”
Line 125: “those of ostracods”
Line 134: “…indicated by Uffernorde [54]…”
Lines 138-141: “… three metrics were calculated: species richness (S), total density (i.e. the number of individuals in each sample), and the Shannon-Weaver diversity index (H).” Move Table 1 citation in the results.
Lines 147-148: “The presence/absence data… were also included for comparative purposes…”. This sentence is not clear. The SIMPER analysis between groups resulting from the cluster analysis was not mentioned.
Table 1: Correct the caption considering what suggested for lines 138-141 and reduce the table dimension to one page.
Lines 159 and 167: “predicted location”?
Line 163: “…methods, such as Kringing, IDW…”
Line 181: “… reported by Grasshoff [63]…”
Lines 187-189: Include here the abbreviations (e.g. dissolved oxygen - DO)
Line 198: “(Dissolved Inorganic Nitrogen, DIN, as sum..)”
Line 206: “while Masoli [46]…”
Line 211: “In the same area, Masoli [46,58]…”
Line 214: “between 0… and 2.7…”
Line 230: “were recorded”
Table 2: Reduce the table dimension including only the main species (e.g. up to 90% cumulative contribution) and better specify the column labels (e.g. what are clusters 1, 2, …? Mean density?)
Line 254: “DO, expressed…, whereas Chl a, which…”
Line 258: “…reported for the period…”
Line 260: “of DIN.”
Line 261: “it accounted for”
Lines 271-274: Move to Materials and Methods.
Lines 283-287: Move to Materials and Methods.
Table 3: “statistics”, “percentile”, “N-NO2-“, …
Figure 3 caption: “between chemico-physical variables”
Line 297: “of growing urbanization”
Line 307: “Yashuara et al. [5] and Wilkinson et al. [43]…”
Line 315: “discharges”
Line 324: “rainfalls”
Line 328: “20th”
Line 357: “opportunistic and hypoxia-tolerant species”
Lines 358-359: Remove this sentence and move the references above.
Line 359: “In addition, Salvi et al. [38]…”
Line 361: “…Harbor. Both species …”
Line 363: “…reported by Masoli [46,58]…”
Line 370: “over the last”
Figure 5: What does this figure show? Specify the association between each image and species in the caption or increase the font size in the legend.
Lines 402-403: Remove this sentence, it is too specific for the conclusions.
Line 407: “records”
Line 410: “These data are in agreement…”
Lines 419-421: Remove this sentence or move it to the Discussion section, it is too specific for the conclusions. Avoid the use of references in the Conclusions section.
Plate A1: “(Masoli, 1968)”. Remove references to scale bars from the caption.
Author Response
Authors: kind reviewer thank you for the useful suggestions and corrections. Here is our comments and answers to your report. We also point out that the new parts in the text are inserted in yellow color.
Authors: All revisions have been accepted and corrected with the exception of the part concerning the chemical-physical results. We believe it is more correct not to move the signaled sentences in the materials and methods.
Reviewer 3 Report
Thank you for submitting your manuscript to the Sustainability journal. Generally, the manuscript fits into the scope of the journal. However, there are some more comments that require revision.
In the literature review, it is important that the scientific novelty of the work is established through a critical analysis of related literature. References from grey literature and non-peer reviewed sources should be avoided. The literature research must be improved substantially. How does this work contribute towards the gaps identified? How does it improve upon previous work? Please avoid clustered references! It is recommended that a short discussion of the novel contribution of each reference cited be provided to give readers a better understanding of their relevance. Further, the scope of the manuscript must be clearly defined.
The methodology must be improved. I strongly recommend to include a flow chart illustrating the steps of the methodology.
In the results section, the graphs in figure 3 and 5 are blury and need to be revised. It should be indicated the source of all figures (also in case they are produced by the authors). The map source must be provided.
In the conclusions, in addition to summarising the actions taken and results, please strengthen the explanation of their significance. It is recommended to use quantitative reasoning comparing with appropriate benchmarks, especially those stemming from previous work.Please elaborate in more detail the conclusions regarding urban environment and biodiversity loss.
Author Response
Authors: Kind reviewer thank you for the useful suggestions and corrections. Here is our comments and answers to your report. We also point out that the new parts in the text are inserted in yellow color.
In the literature review, it is important that the scientific novelty of the work is established through a critical analysis of related literature. References from grey literature and non-peer reviewed sources should be avoided. The literature research must be improved substantially. How does this work contribute towards the gaps identified? How does it improve upon previous work? Please avoid clustered references! It is recommended that a short discussion of the novel contribution of each reference cited be provided to give readers a better understanding of their relevance. Further, the scope of the manuscript must be clearly defined.
Authors: The basis for the taxonomy of the Adriatic ostracods are represented by volumes and publications belonging to the grey literature. Since this is the first work carried out on ostracods in the innermost area of the GoT, there are no previous works that can be relied upon as a term of comparison. In the examined area, moreover, there are few works analyzing the relationship between anthropization and evolution of benthic organisms (see foraminifera, mollusks). Most of the works are related to the evolution of organisms in relation to the anthropic impact in sedimentary series (cores) which pertaining, because of the sampling methodology, to the time span much wider (500-1500 years) than the period we considered.
Authors: A short discussion of the novel contribution of each reference cited was provided. The literature research was improved substantially underlining how the work contribute towards the gaps identified further by better defining the scope of the manuscript
The methodology must be improved. I strongly recommend to include a flow chart illustrating the steps of the methodology.
Authors: Flow chart to improve the methodology was added;
In the results section, the graphs in figure 3 and 5 are blurry and need to be revised. It should be indicated the source of all figures (also in case they are produced by the authors). The map source must be provided.
Authors: The graphs have been revised. The source of all figures has been indicated such as the map source;
In the conclusions, in addition to summarizing the actions taken and results, please strengthen the explanation of their significance. It is recommended to use quantitative reasoning comparing with appropriate benchmarks, especially those stemming from previous work. Please elaborate in more detail the conclusions regarding urban environment and biodiversity loss.
Authors: In the Discussion were summarized the actions taken and results, strengthening the explanation of their significance with the addition of other more recent bibliographical references. Conclusions regarding urban environment and biodiversity loss was elaborate with more detail.
Authors: The text has been revised and corrected by the English language expert of the Department of Mathematics and Geosciences dr. Karry Close
Round 2
Reviewer 1 Report
I think the article can be published in this form.
Reviewer 3 Report
Thank you for the revision. My comments have been considered.